# Radiation Implication in Pediatric Second Primary Thyroid Malignancy (SPTM) Cumulative Incidence and Mortality in the United States: Large Cohort Evidence

**DOI:** 10.3390/cancers16213637

**Published:** 2024-10-29

**Authors:** Laurens Holmes, Casey Lu Simon-Plumb, Ruth Ziegler, Benjamin Ogundele, Mackenzie Holmes, Kirk Dabney, Maura Poleon, Michael Enwere

**Affiliations:** 1Global Health Equity Foundation, GHEF, Bear, DE 19701, USAbenjaminogundele@gmail.com (B.O.); mikky89@gmail.com (M.E.); 2Biological Sciences Department, University of Delaware, Newark, DE 19716, USA; 3Public Health & Allied Health Sciences Department, Delaware State University, Dover, DE 19901, USA; 4Swarthmore College, Swarthmore, PA 19081, USA; 5Biomedical Sciences, Texas A & M University, College Station, TX 77843, USA; holmesmackenzie04@gmail.com; 6Nemours Children Health, Wilmington, DE 19803, USA; kirk.dabney@nemours.org; 7Miami Baptist Hospital, Miami, FL 33176, USA; maurapoleon@gmail.com

**Keywords:** pediatric thyroid cancer, second primary thyroid malignancy (STPM), radiation and thyroid cancer, second primary thyroid cancer incidence and race, health disparities

## Abstract

The purpose of this pediatric cancer research investigation was to assess how radiation therpay in the first primary malignannt neoplasm predispose to Second Primary Thyroid Maligancy (SPTM), as well as therapeutics and mortality associated with SPTM. This study applied a retrospective cohort design, as non-experiment translational epidemiologic investigation, utilizing the USA National Cancer Institute, Surveillance Epidemiology and End Results (SEER) data in the assessment of risk determinants and predisposing factors in SPTM, as well as racial differentials in SPTM cumulative incidence and mortality. Specifically, radiation utilization in first primary malignancy among children predisposed to SPTM, indicative of the need to assess exposure effect of other carcinogens prior to radiation therapy in first primary malignant neoplasm, hence SPTM marginilaization.

## 1. Introduction

During 2022, US children diagnosed with malignant neoplasm were observed with an estimated 18.1 million survival. Specifically, during 2020, an estimated 496,000 children were diagnosed with malignant neoplasia, and followed for the disease, with an estimated 85% survival [1]. Over the past three decades, the pediatric cancer survival optimized, due to novel and reliable therapeutics. Despite an increase in the incidence of childhood cancer, the 5-year survival rates currently exceed 70–80%, [2] indicative of survival enhancement. However, there are limited data on second primary pediatric malignancy (SPPM), including pediatric thyroid cancer (STPM), considered rare among children and young adults, with an estimated 2.0% of diagnoses during the past two decades [2]. Despite the late stage of pediatric thyroid cancer, the prognosis remains improved among children and young adults, with mortality rates of less than 2.0% [3]. Regardless of the observed survival advantage, thyroid cancer investigations continue to present an opportunity for further risk characterization, given several predisposing and causal factors. Available epidemiologic data have observed an increased incidence of thyroid cancer in the last thirty years, with substantial annual increases commencing in the early 2000s [4,5,6,7]. The increasing rates were observed in papillary thyroid cancer (PTC), the most common histological type accounting for 80% of thyroid malignancies [4,5,8].

With the relatively uncommon occurrence of malignant thyroid neoplasm, randomized trials and assessments for reliable therapeutics and value care had not been adequately assessed. However, treatment has relied on standardized treatment typically consisting of a thyroidectomy followed by radioactive iodine (RAI) therapy [5]. The association between radiation exposure during childhood and thyroid cancer development was first elucidated in the 1950s and 1960s. This was due to a combination of radiation being a recognized carcinogen and the radiosensitivity of the thyroid gland [2]. Applying this to the pediatric population is of particular concern because of children’s increased sensitivity to radiation of the cells [3], resulting in retinoblastoma protein (pRB) and cyclins D and E dysregulation and overexpression. The cell cycle is driven by cyclins, namely A, B, D, and E, each present at a particular phase of the cell cycle. The cyclins D mediate between the proliferative pathways and the core cell cycle machinery. The pRB is able to be localized where DNA breaks during the repair process and to assist in non-homologous end joining and homologous recombination. Once at the breaks, pRB is able to recruit regulators of chromatin structure such as the DNA helicase transcription activator. Further, pRB recruits protein complexes such as condensin and cohesin which facilitate the structural maintenance of chromatin.

Additionally, this radiation exposure may result in defective Fos and Jun proteins as well as MYC overexpression, impairing the transcription factors and resulting in abnormal cellular proliferation as neoplastic cells. The ongoing epidemiologic data implicating radiation in thyroid cancer are supported by increased incidence trends of thyroid cancer in children compared to adults following historic nuclear accidents like Chernobyl [5].

The risk of developing a second primary thyroid malignancy (SPTM) requires accurate and reliable studies on this radiation exposure effect in thyroid malignancy. A survey of young adult cancer mortality observed a 16.7% mortality associated with thyroid cancer and 21.4% mortality from subsequent cancer [6]. Currently, most research on the excess risk of SPM development due to radiation has been in adult populations [2]. The relationship between radiation intensity and the risk of developing thyroid cancer observed a direct linear correlation until 20 Gy, then declining while remaining elevated. Studies that have described this association demonstrated that age differentials (higher risk at younger ages) and sex differentials (higher risk for females) modified dose-related risk [2,9].

The survival study by Keegan et al. suggests that race, mainly African American and Hispanic descent, may influence thyroid cancer survival [5]. Further epidemiologic investigation of the social determinants in SPTM incidence, prognosis, and survival is required. We sought in the current study to characterize childhood SPTM by the health disparities domain and to examine the effect of radiation exposure on STMP incidence and mortality, if feasible, using the Surveillance, Epidemiology, and End Results (SEER) dataset from the National Cancer Institute (NCI). We postulated that the widespread use of neuro-diagnostic imaging, especially in pediatric settings, as well as radiation and chemotherapy for first primary tumors in childhood, may contribute to increased mutation, epigenomic alterations, and abnormal cellular proliferation, hence SPTM. Furthermore, the relationship between radiation and SPTM may be confounded by race, sex, or age at diagnosis, thus creating potentially biased estimates and the heterogeneity effect for health inequities and social determinants of health.

## 2. Materials and Methods

The data use approval (DUA) from the National Cancer Institute (NCI), a National Institute of Health (NIH) agency, was applicable in this study. This DUA involved utilizing the Surveillance Epidemiology and End Results (SEER) dataset, SEERStat, 8.4.3.

### 2.1. Study Design

A retrospective cohort study (case-only) was used to characterize the temporal trends in SPTM and to determine the exposure effect of radiation in the predisposition to SPTM. Whereas cohort studies, mainly prospective, depend on the pre-classification of exposure before the study’s commencement, a retrospective cohort design attempts to determine the outcome based on the exposure history. A retrospective cohort design reflects an epidemiologic non-experimental survey that examines the exposed compared to the non-exposed, implying radiation as an exposure. This non-experimental design is appropriate, given the nature of the data as pre-existing and secondary. The design used in this study, a case-only retrospective cohort, is adequate given that radiation is the exposure. The binary distribution among the SPTM cases allows for its exposure to function as the cause of SPTM. However, like all retrospective designs, this design exercises caution in addressing reverse causation. The design perspective of this study was based on the nature of the data, implying secondary or pre-existing data. Since the exposure variable, mainly radiation, was measured on a binary scale, a binomial regression model was utilized. This model allows for a reliable outcomes prediction: second primary thyroid malignancy and mortality. The binomial regression reflects a generalized linear model implying the risk in the exposed, such as radiation, and the risk in the unexposed, mainly non-radiation, and the computation of the risk estimate based on the risk in the exposed divided by the risk in the unexposed. This predictive model estimated the effect of radiation on second primary thyroid malignancy. The modeling approach used in this study, which reflects a secondary data analysis, is indicative of pre-existing data. In this modeling, we examined the variables to ensure their accuracy before modeling.

### 2.2. Data Source: SEER Registry, Study Population, and SPTM Sample

The SEER database is the NCI tumor registry for epidemiologic and health research, such as diagnosis and treatment received. This database comprises population-based tumor registries that routinely collect information on all newly diagnosed cancer cases confirmed through histopathology in SEER-eligible cancer registries. The database includes patients’ sociodemographic information, age at diagnosis, tumor stage and grade, tumor type, tumor markers, surgery, the radiation received, vital status, and cause-specific mortality, among other information. The data used in this analysis involve the National Cancer Institute, using surveillance and epidemiology result data. These data are considered reliable in risk estimations, survival, and mortality.

The nine original registries (not including the Seattle and Atlanta Metropolitan registries that joined in 1975) represent 11% of the United States population. These registries are recognized by the North American Association of Cancer Registries for the highest level of data quality certification. This dataset commenced data acquisition from several states in the United States in 1973 based on the Italian Institute of Tumor data appraisal and the incorporation of their principles in the SEER dataset. This SEER registry indicates a representative sample of cancer data in the United States. Additionally, before the individual states in the United States became a part of this registry, reliable criteria were required, implying accurate data submission without misclassification from these states’ cancer registries.

The variables utilized in this study were examined based on the scale of measurement, namely qualitative and quantitative scales. This approach allows for accurate and reliable modeling within the regression.

### 2.3. Study Eligibility

The study eligibility indicates inclusion and exclusion criteria in this modeling using a retrospective cohort design. The inclusion criteria are as follows: (a) second thyroid malignancy, (b) primary thyroid malignancy, (c) children 0–19 years, (d) study duration, and (e) ICD10 nosology. The exclusion criteria are as follows: (a) children with malignancy in the SEER registry diagnosed with other malignancies besides primary and second primary thyroid malignancies; (b) children with second primary thyroid malignancy diagnosed before 1973 due to International Classification of Disease (ICD), Version 10 nosology criteria variance; (c) if the cases, namely first and second primary thyroid malignancy, were observed without tumor grade, radiation therapy variable, tumor primaries, and tumor staging; and (d) children older than 19 years. These cases were excluded from the analysis.

### 2.4. Study Sample

In this retrospective cohort study case-only, comparing children who received radiation therapy versus no radiation therapy, the study population (*n* = 3518) comprised children diagnosed with thyroid cancer between 1973 and 2014, aged 0–19 years in the SEER registry. All races, except unknown (*n* = 61), were included in the study that met the inclusion criteria of age at diagnosis. Ultimately, 3457 children were assessed as the study population, while the study sample compromised all children with SPTM (*n* = 99).

### 2.5. Statistical Analyses

A pre-analysis screening was performed, implying the data examination to identify missing variables, errors, and outliers using a box plot, stem, and leaf. If the data met the normality assumption, the mean and standard deviation (SD) were used to summarize the quantitative scale of data measurement as a continuum; otherwise, the median and interquartile ranges were utilized. This variable was summarized using mean and standard deviation. To summarize the qualitative variables, frequency or counts and percentage were applied for categorical scale, mainly race and sex. A chi-square tabulation analysis was used to assess the sex and race distribution of STMP and radiation.

The hypothesis-driven analysis, namely the effect of radiation on second primary thyroid malignancy, was estimated using a binomial regression model. The margins plot was used to examine the annual percent change and the percent change with respect to cumulative thyroid cancer incidence. A unimodal binomial regression model was utilized with radiation as the single independent or predator variable in the model. The same statistical modeling approach was applied to mortality estimation. Since a single predictor as independent and explanatory variable cannot fully explain the outcome, such as second primary thyroid malignancy among children, a confounding assessment was applied, implying a magnitude of confounding greater than 10% prior to incorporation into the multivariable model. This approach as confounding assessment and magnitude estimation allows for the adjustment of confoundings prior to the risk estimation in radiation as the exposure effect in second primary thyroid malignancy as well as in mortality.

The first model in the multivariable analysis utilized radiation as a single variable, while the next level of modeling incorporates demographics such as sex and race. Finally, the subsequent modeling utilized radiation, demographics, and tumor prognostic factors in the model building.

A two-tail test was used in the hypothesis testing, implying no radiation risk is associated with second primary thyroid malignancy. To quantify the random error, implying the *p*-value, the type 1 error tolerance was 5% (0.05), while the confidence interval as the measure of precision was 95%. In the multivariable modeling, the *p*-value was 1% (0.01), while the confidence interval was 99%. The entire analyses were performed using STAT (version 15, STATA Corporation, College Station, TX, USA).

### 2.6. Percent Change and Annual Percent Change Trends Analysis

To examine the age-adjusted incidence trends for SPTM, a weighted mean average was used with the 2000 United States standard population as the denominator. The Percent Change (PC) was calculated using one year for each endpoint, while Annual Percent Change (APC) was estimated using the Weighted Least Squares method. This method utilized random error quantification in the model, implying sampling variability assessment as *p* value, as well as the application of regression functions that are either linear or nonlinear in the parameter estimates of the sample statistic. The weight size illustrates the precision of the information in the associated observation, implying the parameter value and precision. Therefore, optimizing the weighted fitting measure to identify or examine the parameter estimates allows the weights to determine the contribution of each observation to the final parameter estimates. The APC was used over several years to measure age-adjusted incidence trends from 1973 to 2014. Confidence intervals for both parameters, mainly PC and APC, were set at 95%. The entire analyses were performed using SEER*Stat 8.3.4 statistical software [10].

## 3. Results

### 3.1. Study Characteristics

Although not shown on a table, there were 3457 pediatric thyroid cancer cases and 99 (2.8%) were diagnosed with SPTM. The racial distribution of the sample reflected whites (*n* = 2990, 86.5%), blacks (*n* = 173, 5.0%), and others such as American Indian/Alaska Native and Asian/Pacific Islander (*n* = 294, 8.5%). Most of the cases were female (*n* = 2791, 80.7%) and were in the age group of 15–19 years old (*n* = 2576, 74.5%). The tumor grade was observed as unknown tumor grade classification (*n* = 2673, 77.3%), well differentiated (Grade I), *n* = 589, 17.0%. The cumulative incidence (CmI)) of mortality was 0.13, (130, per 1000).

The study characteristics of children diagnosed with SPTM are presented in Table 1. Between 1973 and 2014, there were 99 children diagnosed with SPTM and followed for the disease.

The age, sex, race, mortality, tumor grade, and insurance are described during this period of diagnosis. Second primary thyroid malignancy (SPTM) was mainly diagnosed among the ages 10–14 and 15–19. With respect to sex, there were more females diagnosed with SPTM (60.6%). Racial differences were observed in SPTM diagnosis, with most cases among whites (88.9%). Relative to other childhood malignancies, SPTM demonstrated a survival advantage (86.9%), implying that 13.1% of the children diagnosed with SPTM experienced mortality. The majority of SPTM cases were well differentiated (17.2%). Concerning insurance coverage, an estimated 44% had health insurance.

### 3.2. Cumulative Incidence Trend of SPTM

The PC for SPTM from 1973 to 2014 was 151.2 for all races and 99.8 for whites. Table 2 demonstrates the age-adjusted incidence rates of SPTM in children aged 0–19 years by race and sex. Figure 1 illustrates the age-adjusted incidence rates of SPTM in children aged 0–19 years by race. Despite the spike in cumulative incidence between 1993 and 1997, there was a steady increase in a cumulative trend by whites during the entire period. Figure 2 illustrates the age-adjusted incidence rates of SPTM in children aged 0–19 years by sex. Despite cumulative incidence rate differences, the trends were comparable in males and females, indicating an increase from 1973 through 2014. The cumulative incidence rates were estimated with 5-year intervals from 1973 to 2014, except years 2012–2014, which was only a two-year interval.

The highest cumulative incidence rate during the study period (1973–2014) was observed among whites (0.016, 95% CI 0.000–0.536 per 100,000 and age-adjusted to the 2000 United States Standard population) and others (0.016, 95% CI 0.000–3.808), while blacks (0.005, 95% CI 0.000–2.754) had the lowest cumulative incidence rate. The cumulative incidence rate was higher in females (0.015, 95% CI 0.000–0.813) compared to males (0.010, 95% CI 0.000–0.774).

### 3.3. Exposure Effect of Radiation, Sociodemographics, and Geographic Locale of SPTM

The exposure effect of radiation on SPTM is presented in Table 3. There was no statistically significant association between radiation and SPTM risk, risk ratio (RR.) = 1.03, 95% CI 0.69–1.52. Relative to males, females were 63% less likely to develop SPTM, RR = 0.37, 95% CI 0.25–0.54. There was a dose-response in the development of SPTM, comparing geographic locale, namely, metropolitan, rural, and urban. Compared to rural regions, children in the urban areas were 20% less likely to develop SPTM, RR = 0.80, 95% CI 0.18–3.51, while those in the metropolitan areas were 38% less likely to receive radiation.

### 3.4. Multivariable Modeling: Radiation and SPTM Risk Estimation

The adjustment for factors considered confounding to the effect of radiation on SPTM is illustrated in Table 4. After controlling for the effect of sex, tumor grade, race, and geographic locale, on the association between radiation therapy received and the development of SPTM, radiation was associated with a 5% increased likelihood of SPTM, adjusted risk ratio (aRR) = 1.05 99% CI 0.63–1.75. In the same model, although statistically insignificant, blacks were 60% less likely to develop SPTM, aRR = 0.40, 99% CI, 0.06–2.47, while others were 18% more likely to develop SPTM, aRR = 1.18, 99% CI, 0.48–2.87. Likewise, females compared to males were 63% less likely to develop SPTM, aRR = 0.37 99% CI 0.22–0.61. Relative to rural areas, SPTM was less diagnosed in urban areas (aRR = 0.79, 99% CI, 0.12–5.35) and metropolitan areas (aRR = 0.60, 99% CI 0.10–3.59).

## 4. Discussion

SPTM has been increasing during the past two decades in most populations and in the pediatric setting. The increasing trends may be due to advances in radiographic imaging/diagnostics. While there are few studies with limited sample sizes in adult environments, there are no studies to our knowledge in the pediatric environment, and neither are there studies on risk characterization in the development of SPTM. The current study was proposed to examine temporal trends in SPTM between 1973 and 2014 using the United States NCI SEER data, as well as the exposure effect of radiation on SPTM risk. There are a few relevant findings from this study. First, the cumulative incidence of SPTM indicates an increasing trend from 1973 to 2014. Secondly, regardless of sex, there was a steady increasing trend, albeit with incidence rate differences. Thirdly, the cumulative incidence and incidence rate were higher among whites relative to blacks. Fourthly, a marginal increase in the SPTM cumulative incidence was associated with radiation. Fifthly, relative to whites, blacks were less likely to be diagnosed with SPTM. Finally, SPTM was more diagnosed among children in rural and urban regions compared to metropolitan areas.

This finding observed the increasing cumulative incidence trends of SPTM in the United States [7,11]. These data or findings are supported by previous data on increased second primary malignancy in the adult population [6,11,12,13]. The observed trend in our sample may be due to survival improvement across all types of malignancies, implying an increased risk of developing SPTM following treatments involving radiation therapy and other oncologic agents. These novel oncologic agents may result in the expression of several biomarkers, the dysregulation of apoptosis and tumor suppressor genes, and cyclin D elaboration. Additionally, the increasing trend in SPTM may be due to advances in tumor diagnosis or changes in tumor nosology (disease classification) [7]. Furthermore, unknown but potential carcinogens associated with thyroid tissue proliferation may explain the increase in SPTM incidence.

The current findings in this study demonstrate an increase in SPTM incidence as well as cumulative incidence despite the incidence rate difference. However, this direction is not backed by any epidemiologic, clinical, or translational data. Overall, cancer has been associated with an increased incidence among males and poorer survival [6,14]. The observed cumulative incidence among females with respect to SPTM may be due to the poorer survival of males, consistent with an epidemiologic curve in mortality events. This curve simply reflects increasing events (disease incidence and mortality) as a function of virulence, pathogenicity, and population size.

In our sample, blacks relative to whites were less likely to be diagnosed with SPTM. This observation supports previous data on the incidence of major pediatric malignancy, namely acute lymphoblastic leukemia (ALL), brain/central nervous system (CNS) tumor, and primary thyroid malignancy [6,15]. Race is often a robust proxy for class, encapsulating access and ability to afford advanced diagnostic and therapeutic procedures. The increased risk of SPTM in whites may reflect an increased frequency and exposure to radiation diagnostic imaging and therapeutics due to accessibility or earlier cancer detection, resulting in a longer radiation exposure time frame. While blacks tend to be less diagnosed with pediatric tumors in general, the lower cumulative incidence of SPTM in our sample may be due to the survival disadvantage of pediatric malignancy among black children [6,7,14,16,17]. 

Additionally, the lower cumulative incidence among blacks may reflect a lack of access to and utilization of cancer treatment and preventive services [6,17,18,19,20,21].

We have observed that radiation was associated with SPTM in our sample. Previous studies have implicated radiation exposure in thyroid cancer development in adults and children [2,22,23,24]. The observed association in our sample may be causal, given the biologic mechanism and the mechanistic role of radiation exposure and abnormal cellular differentiation (malignant neoplasm). Radiation remains mutagenic with a proliferative role in tumor predisposition and risk. The thyroid gland is vulnerable to radiation, given its anatomic position, implying the likelihood of being irradiated and the ability to concentrate iodine [3]. Additionally, rapidly growing tissues, as observed in early developmental stages (pediatric environment), are more vulnerable to mutagenic changes [23].

Both urban and rural areas compared to metropolitan were associated with increased cumulative incidence and SPTM risk. The observed increased trend and diagnosis in rural and urban areas compared to the metropolitan may be due to geo-racial implications of risk in cancer and other disease conditions. Available epidemiologic data observed rural cases with later cancer stages at diagnosis and worse prognoses. This variance in “rural disadvantage” may imply more aggressive treatment and/or radiation therapy and thus corresponds to an increased risk for SPTM development [25]. In this study, a greater proportion of blacks reside in metropolitan areas than whites, which may explain the excess SPTM diagnosis in rural and urban areas. Our data are supported by previous findings in adult malignancies that tend to segregate around the geographic locale and physical environment [26]. However, while radiation is implicated in second primary thyroid malignancy among children, there are no data for assessing the genomic and epigenomic alterations or aberration and the implication in SPTM. The assessment of the genomic and epigenomic data may provide more useful information on the implication of radiation in second primary thyroid malignancy among children, thus elucidating the gene and environment interaction in transcriptome dysregulation, impaired gene expression, and cellular dysfunctionality.

Despite the strengths of this study, namely, the large sample size of pediatric SPTM, appropriate study design, and statistical modeling, there are some limitations. First, these findings may be driven by unmeasured confounding since the dataset does not provide information on substantial determinants of health and health inequities resources. The collaboration required in this field today (radiation and gene interaction) involves the knowledge of epigenomic aberrations resulting in impaired gene expression and protein synthesis dysfunctionality. In effect, the racial variance in SPTM may be explained by DNA hypomethylation and hypermethylation that result in impaired gene expression and abnormal cellular proliferation of the thyroid gland. Specifically, black children who survive the first primary malignancy may differ in the DNA methylation that maps the pathway for proliferative proteins and SPTM development. Secondly, while radiation has been observed in thyroid malignancy, and in other malignancies such as brain and oral cavity neoplasm, this study reflects an insufficiency due to the age group of the patients in this evaluation. Due to this observation, it is improbable that the observed risk of radiation in second primary malignancy remains non-inferential, due to sampling variability. Thirdly, despite controlling for confounding in the association between radiation and SPTM, these findings may be driven by residual confounding, since no matter how sophisticated the statistical software that is used in adjusting for confounding, residual confounding remains [27,28,29,30]. Nevertheless, it is implausible that our findings of the association between radiation therapy received and other risk markers of SPTM are solely driven by unmeasured and residual confounding [31,32,33,34]. Lastly, the inability of NCI to provide data on the primary therapeutic, standard of care, and other prognostic factors that allow for an explanatory model if incorporated into the analysis is confounding, requiring the availability of these data for a reliable explanatory model.

## 5. Conclusions

In summary, the cumulative incidence of SPTM continues to increase and varies by sex, race, and geographic locale. In addition, SPTM is marginally associated with radiation in terms of risk characterization. These findings indicate the need to embrace multiple risk variables in understanding the mechanistic pathway in SPTM to inform clinical medicine and public health on the implication of radiation in both diagnostic and therapeutic settings, reducing the incidence of SPTM in the pediatric population, especially among whites.

## Figures and Tables

**Figure 1 cancers-16-03637-f001:**
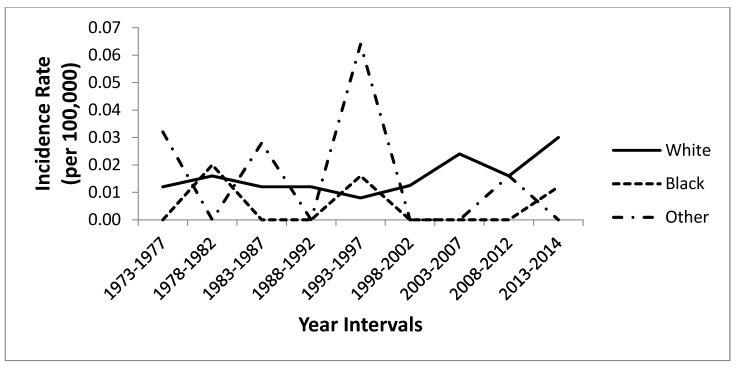
Age-adjusted incidence rates of pediatric SPTM (per 100,000) by race, Surveillance, Epidemiology, and End Results (SEER), 1973–2014. Notes: Age-adjusted incidence rates of children with SPTM by race.

**Figure 2 cancers-16-03637-f002:**
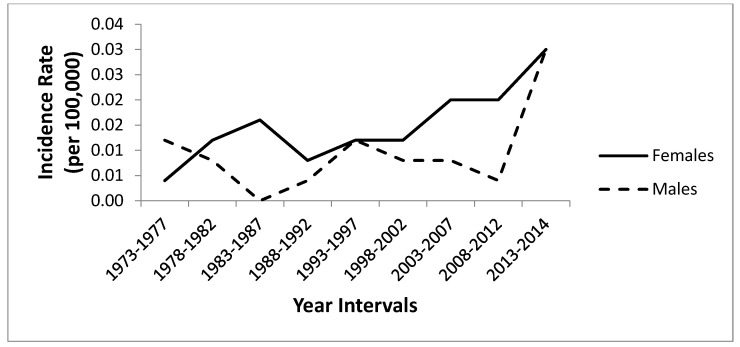
Age-adjusted incidence rates of pediatric SPTM (per 100,000) by sex, Surveillance, Epidemiology, and End Results (SEER), 1973–2014. Notes: Age-adjusted incidence rates of children with SPTM by sex.

**Table 1 cancers-16-03637-t001:** Study characteristics of subjects with second primary thyroid malignancy (SPTM), SEER, 1973–2014 (SPTM were diagnosed after primary thyroid cancer).

Variable	*n*	%
Age group		
<1 year	0	0
01–04 years	0	0
05–09 years	3	3
10–14 years	25	25.3
15–19 years	71	71.7
Sex		
Male	39	39.4
Female	60	60.6
Race		
White	88	88.9
Black	2	2
Other (AI/AN, AS/PI)	9	9
Mortality Status		
Dead	13	13.1
Alive	86	86.9
Tumor grade		
Well differentiated; Grade I	17	17.2
Moderately differentiated; Grade II	2	2
Poorly differentiated; Grade III	0	0
Undifferentiated; Anaplastic; Grade IV	0	0
Unknown	80	80.8
Insurance type		
Medicaid (public)	17	17.2
Insured (private/commercial)	23	23.2
Insured/No specifics	4	4
Uninsured	1	1
Unknown	0	0
Blanks	54	54.6

**Notes and Abbreviations:** SPTM cases were diagnosed after primary thyroid cancer; SEER = Surveillance Epidemiology, and End Results; AI/AN = American Indian/Alaska Native’ AS/PI = Asian/Pacific Islander; *n* = counts or frequency; % = percentage.

**Table 2 cancers-16-03637-t002:** Age-adjusted incidence rates of SPTM (per 100,000) by race and sex of children, Surveillance, Epidemiology, and End Results Program, 1973–2014 (SPTM cases were diagnosed after primary thyroid cancer).

Variable	Five Year Interval								
	1973–‘77	‘78–‘82	‘83–‘87	‘88–‘92	‘93–‘97	‘98–2002	‘03–‘07	‘08–‘12	‘13–‘14
	Rate (95% CI)	Rate (95% CI)	Rate (95% CI)	Rate (95% CI)	Rate (95% CI)	Rate (95% CI)	Rate (95% CI)	Rate (95% CI)	Rate (95% CI)
Race									
White	0.012	0.016	0.012	0.012	0.008	0.013	0.024	0.016	0.03
	(0.000–0.556)	(0.000–0.536)	(0.000–0.520)	(0.000–0.516)	(0.000–0.520)	(0.000–0.528)	(0.000–0.528)	(0.000–0.536)	(0.000–0.580)
Black	0	0.02	0	0	0.016	0	0	0	0.012
	(0.000–3.964)	(0.000–3.388)	(0.000–3.152)	(0.000–2.748)	(0.000–2.680)	(0.000–2.476)	(0.000–2.412)	(0.000–2.284)	(0.000–1.683)
Other (AI/AN, AS/PI)	0.032	0	0.028	0	0.064	0	0	0.016	0
	(0.000–6.572)	(0.000–5.196)	(0.000–4.412)	(0.000–3.780)	(0.000–3.528)	(0.000–3.116)	(0.000–2.708)	(0.000–2.536)	(0.000–2.420)
Sex									
Male	0.012	0.008	0	0.004	0.012	0.008	0.008	0.004	0.03
	(0.000–0.904)	(0.000–0.812)	(0.000–0.764)	(0.000–0.752)	(0.000–0.760)	(0.000–0.744)	(0.000–0.728)	(0.000–0.720)	(0.000–0.780)
Female	0.004	0.012	0.016	0.008	0.012	0.012	0.02	0.02	0.03
	(0.000–0.924)	(0.000–0.848)	(0.000–0.864)	(0.000–0.788)	(0.000–0.788)	(0.000–0.768)	(0.004–0.772)	(0.000–0.768)	(0.000–0.800)

**Notes and Abbreviations:** SPTM cases were diagnosed after primary thyroid cancer AI/AN = American Indian/Alaska Native; AS/PI = Asian/Pacific Islander; CI = Confidence Interval. These findings indicate a five year interval in age-adjusted incidence rate with 95% CI.

**Table 3 cancers-16-03637-t003:** Exposure effect of radiation and other variables on SPTM in univariable model, SEER (1973–2014).

Variable	Risk Ratio	95% CI	*p*-Value
Radiation			
No	1	Referent	Referent
Yes	1.03	0.69–1.52	0.9
Sex			
Male	1	Referent	Referent
Female	0.37	0.25–0.54	<0.001
Race			
White	1	Referent	Referent
Black	0.4	0.06–2.47	0.19
Other (AI/AN, AS/PI)	1.18	0.48–2.87	0.64
Tumor grade			
Well differentiated; Grade I	1	Referent	Referent
Moderately differentiated; Grade II	0.5	0.12–2.13	0.35
Unknown	1.04	0.62–1.74	0.89
Urbanicity (geographic locale)			
Rural	1	Referent	Referent
Urban	0.8	0.18–3.51	0.77
Metropolitan	0.62	0.16–2.45	0.5

**Abbreviations and Notes:** SPTM= secondary primary thyroid malignancy; AI/AN = American Indian/Alaska Native, AS/PI = Asian/Pacific Islander; CI = Confidence Interval. The type I error tolerance (*p* value) was set at 5% (0.05).

**Table 4 cancers-16-03637-t004:** Exposure effect of radiation and other variables on SPTM in a multivariable model, SEER (1973–2014).

Variable	Adjusted	99% CI	*p*-Value
	Risk Ratio		
Radiation			
No	1	Referent	Referent
Yes	1.05	0.63–1.75	0.82
Sex			
Male	1	Referent	Referent
Female	0.37	0.22–0.61	<0.001
Race			
White	1	Referent	Referent
Black	0.4	0.06–2.47	0.19
Other (AI/AN, AS/PI)	1.18	0.48–2.87	0.64
Tumor grade			
Well differentiated; Grade I	1	Referent	Referent
Moderately differentiated; Grade II	0.52	0.08–3.55	0.39
Unknown	1.02	0.52–2.01	0.94
Urbanicity (geographic locale)			
Rural	1	Referent	Referent
Urban	0.79	0.12–5.35	0.75
Metropolitan	0.6	0.10–3.59	0.47

**Abbreviations and Notes**: SPTM = secondary primary thyroid malignancy; AAI/AN = American Indian/Alaska Native, AS/PI = Asian/Pacific Islander; CI = Confidence Interval. The type I error tolerance was set at 1% (0.01).

## Data Availability

These data, from the SEER Data Source, was approved by NCI prior to acquisition and application in this study.

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
