# Peer review of "Radiation Implication in Pediatric Second Primary Thyroid Malignancy (SPTM) Cumulative Incidence and Mortality in the United States: Large Cohort Evidence"

_cancers, 2024, doi:10.3390/cancers16213637_

Round 1
Reviewer 1 Report
Comments and Suggestions for Authors
The article titled “Radiation Implication in Pediatric Second Primary Thyroid Malignancy (SPTM) Cumulative Incidence and Mortality in the United States: Large Cohort Evidence” has sufficient background information and is well written, here are some concerns:
1. In line 77, what is the age range for this study? Is there a linear relationship between age and SPTM risk?
2. In line 78, what does the “…modified dose-related risk” mean?
3. In line 214, “…mortality was XX%.”, the exact mortality number is missing.
4. When mentioned that “Racial differences were observed in SPTM diagnosis, with most cases among whites (88.9%)”, the authors should discuss the population distribution ratios of whites and blacks in the surveyed areas. Is this result due to the high proportion of whites in the surveyed areas?
5. In line 231-240, how to generate the age-adjusted incidence rates?
Author Response
Attached please find one's reply

Reviewer 2 Report
Comments and Suggestions for Authors
Dear Author
The only point is providing the Ethical committee and registration code and informed consent
Author Response
Attached, please find one's response!

Reviewer 3 Report
Comments and Suggestions for Authors
Here the authors investigate the racial and sex differences in the incidence and mortality of second primary thyroid malignancy (SPTM) in pediatric patients in the United States, focusing on the impact of radiation exposure. Using data from the SEER registry, the study found that the cumulative incidence of pediatric thyroid cancer increased significantly from 1973 to 2014, with SPTM accounting for 3% of these cases. The study highlights that black children were less likely to develop SPTM compared to white children, and females were less likely to be diagnosed with SPTM than males. Additionally, children in urban and metropolitan areas had a lower risk of developing SPTM compared to those in rural areas. The study also noted a slight increase in SPTM risk associated with radiation exposure, although this finding was not statistically significant.
Main points:
The study provides a comprehensive analysis of a large cohort, offering valuable insights into the epidemiology of pediatric SPTM.
It highlights important disparities in incidence and outcomes based on race, sex, and geographic location, which can inform targeted interventions and public health strategies.
The use of a well-established database like SEER adds credibility to the findings.
Minor points:
The study's retrospective design may introduce biases, and the reliance on pre-existing data limits the ability to control for all potential confounding factors.
The findings related to radiation exposure were not statistically significant, which may reduce the impact of this aspect of the study.
Additionally, the study's focus on a specific population may limit the generalizability of the results to other groups.
These points should be discussed in a limitations paragraph in the discussion or conclusion parts.
Author Response
Attached please observe the authors' response.

Reviewer 4 Report
Comments and Suggestions for Authors
Holmes et al., work covers the implications of radiation exposure in pediatric second primary thyroid malignancy (SPTM) in the United States, including racial and sex variances, temporal trends, and mortality rates.
The importance of addressing racial and geographical disparities in cancer diagnosis and treatment is particularly noteworthy. The conclusion could have merit in the related field. Some concerns need to be addressed by the authors.
General comment
There are several areas where clarity, grammatical accuracy, and overall readability could be improved. Some sentences are quite long and complicated, which may hinder reader understanding. Breaking them into shorter, more concise sentences would improve clarity.
Introduction
- Line 43: “the 5-year survival rates 43 currently exceed 70-80%,” Please update the citation for this statistical data and other references at the beginning of the introduction, as it is over 10 years old.
- Line 61: “resulting in pRB, cyclins D and E dysregulation.” Please expand “pRB” at its first mention and add simple note (for example between brackets) regarding the roles of these proteins in the cell for non-specialty future readers.
Methods
- The last date of access the SEER registry by the authors was not clear in the text.
- Why did the authors apply study search until 2014 specifically? This issue was not clear in the text.
- Sampling section (2.4) will be improved by providing a main flowchart illustrates the steps of data collection, the authors applied during their search, including the inclusion/exclusion criteria.
- Please clarify what is meant by “ICD10 nosology criteria.”
- As part of data transparency, the codes used during sample collection should be provided as a supplementary material.
- Line 151: “Children older than 19 years as well as urbanicity.” Individuals > 19 are adults, not children (this criterion was already mentioned in the inclusion criteria). What is the rational of excluding “urbanicity”?
- Line 161: “If the data the normality assumption,…” please revise.
- Line 173: What is mean by “predator variable”?
- The explanation of how the weighted least squares method was used to derive the APC in relation to "random error" could be simplified to improve comprehension.
- Tables 1 and 2 titles “SPTM were diagnosed after primary thyroid cancer).” This information can be better written in the table footer instead of the title.
- Please check the period within which the studies were recruited in this study as it is written (1973-2014) vs. in the inclusion criteria (1973-2013).
Discussion
The discussion section effectively summarized the key findings while addressing the important implications of the study. By integrating clearer language, strengthening connections to the literature, and outlining future research needs, the authors can significantly enhance the impact of this section.
-Thanks to the authors for mentioning the study limitations, in particular, that relate to the “SEER” registry.
Minor comment
When referring to "second primary thyroid malignancy" throughout the text, use a consistent abbreviation (e.g., SPTM) after the first mention for clarity and simplicity.
- Other terminologies with the related acronyms can be revised for the same issue.
Author Response
Attached, please observe the Authors' response.
Thanks

Round 2
Reviewer 1 Report
Comments and Suggestions for Authors
The authors have addressed all my concerns, and this article has been improved sufficiently to be published in Cancers.
Author Response
No Comments
Reviewer 4 Report
Comments and Suggestions for Authors
Thanks for the authors for addressing most of the previously raised concerns.
Still, two essential ones were not addressed.
1- Line 44 and the first paragraph of the introduction section: "The 5-year survival rates currently exceed 70-80%," the cited reference is very old = more than 10 years (not match at all the word "currently." In my previous comments, I asked the authors to update this citation with the related text and the statistical figure according to recent references, as well as update the other citations at the beginning of the introduction (as all >10 years).
2- The sampling section (2.4) will be improved by providing a main flowchart that illustrates the steps of data collection the authors applied during their search, including the inclusion/exclusion criteria. The referee meant a flowchart diagram = Figure to facilitate following the authors in their elaborations.
Author Response
Attached please find the second response.
Thanks for the feedback.
